# Nucleotide, Phospholipid, and Kynurenine Metabolites Are Robustly Associated with COVID-19 Severity and Time of Plasma Sample Collection in a Prospective Cohort Study

**DOI:** 10.3390/ijms25010346

**Published:** 2023-12-26

**Authors:** Haley A. S. Chatelaine, Yulu Chen, John Braisted, Su H. Chu, Qingwen Chen, Meryl Stav, Sofina Begum, Joann Diray-Arce, Jaleal Sanjak, Mengna Huang, Jessica Lasky-Su, Ewy A. Mathé

**Affiliations:** 1Division of Preclinical Innovation, National Center for Advancing Translational Sciences, National Institutes of Health, Rockville, MD 20850, USA; haley.chatelaine@nih.gov (H.A.S.C.);; 2Channing Division of Network Medicine, Brigham and Women’s Hospital, Harvard Medical School, Boston, MA 02115, USA; 3Precision Vaccines Program, Boston Children’s Hospital and Department of Pediatrics, Harvard Medical School, Boston, MA 02115, USA

**Keywords:** longitudinal cohort, regression analysis, COVID-19 severity, untargeted metabolomics, tryptophan metabolism, phospholipid metabolism, biomarkers, pyrimidine metabolism, prospective sampling

## Abstract

Understanding the molecular underpinnings of disease severity and progression in human studies is necessary to develop metabolism-related preventative strategies for severe COVID-19. Metabolites and metabolic pathways that predispose individuals to severe disease are not well understood. In this study, we generated comprehensive plasma metabolomic profiles in >550 patients from the Longitudinal EMR and Omics COVID-19 Cohort. Samples were collected before (*n* = 441), during (*n* = 86), and after (*n* = 82) COVID-19 diagnosis, representing 555 distinct patients, most of which had single timepoints. Regression models adjusted for demographics, risk factors, and comorbidities, were used to determine metabolites associated with predisposition to and/or persistent effects of COVID-19 severity, and metabolite changes that were transient/lingering over the disease course. Sphingolipids/phospholipids were negatively associated with severity and exhibited lingering elevations after disease, while modified nucleotides were positively associated with severity and had lingering decreases after disease. Cytidine and uridine metabolites, which were positively and negatively associated with COVID-19 severity, respectively, were acutely elevated, reflecting the particular importance of pyrimidine metabolism in active COVID-19. This is the first large metabolomics study using COVID-19 plasma samples before, during, and/or after disease. Our results lay the groundwork for identifying putative biomarkers and preventive strategies for severe COVID-19.

## 1. Introduction

In late 2019, coronavirus disease 2019 (COVID-19) started to spread rapidly across the globe. As of January 2023, more than 6.2 million hospitalizations and 1.1 million deaths have been attributed to COVID-19 in the United States [1], with more than three-quarters of a billion total cases and almost 7 million deaths globally [2]. The outbreak led to an explosion of research, vaccine development [3], and COVID-19 treatment therapies [4,5,6,7]. Despite these efforts, the metabolic underpinnings that predispose individuals to severe COVID-19, as well as those that continue to be altered after disease, remain elusive. Such underpinnings lay the foundation required to develop early detection and preventative strategies that healthcare workers can leverage to eliminate or minimize severe disease outcomes.

Metabolomic profiling is a proven approach for developing biomarkers and discovering drug targets for a wide range of diseases, including infectious diseases [8,9]. Specifically, metabolomic profiling yields abundances of small molecules (<1500 Daltons), including lipids, sugars, carbohydrates, nucleotides, amino acids, and drugs/xenobiotics. These measurements are captured in a broad range of biospecimens, such as blood, cells, and urine, and reflect the global metabolic state of a biological system. Plasma metabolomic profiling specifically captures metabolites that reflect systemic metabolism using a minimally invasive blood draw, and is thus a powerful approach for identifying putative clinical biomarkers and molecular pathways/underpinnings. 

In the context of COVID-19, metabolomic profiles can reveal specific metabolites or pathways related to viral infection and/or concomitant host responses [7]. Examples include increased levels of sphingolipids [10] with decreased COVID-19 severity. In other studies, relative changes of lyso-phosphatidylcholines and phosphatidylcholines [11,12], and cytosine- and uridine-related nucleotides [13,14] have also been associated with COVID-19 severity. Many other changes in metabolites and classes, including bile acids [15,16], urea cycle metabolites [14,17,18,19], and metabolites related to stress responses [15,20,21], have been observed in the context of COVID-19 severity in active disease and are listed in Appendix A. 

Notably, all previous studies to date have analyzed samples collected during or after active disease. Further, previous studies are either limited in size (<400 patients), do not account for relevant confounders (e.g., demographics, risk factors, and comorbidities), and/or do not measure a wide range of metabolite species (and therefore do not broadly cover metabolism). Yet, large sample sizes, account of possible confounders, and broad coverage of the metabolome are necessary to provide a clearer and more definitive view of disease-related molecular underpinnings. Further, we note that the timing of sample collection relative to disease diagnosis dictates how metabolomic profiles are interpreted. For example, profiles drawn from samples prior to disease diagnosis yield metabolites that could predispose to disease, while profiles drawn from samples during active disease reflect molecular changes that are most directly associated with the disease itself. Profiles detected in samples after disease can provide important information about potential lingering effects of the disease itself. Importantly, multiple-time-point sampling in the same cohort of individuals enables the discovery of metabolites associated with susceptibility to severe disease, while providing insight into metabolites both initially and persistently associated with COVID-19. Metabolites reflecting persistent metabolic effects of COVID-19 may also uncover biological underpinnings of long-COVID, a potentially greater public health threat than COVID-19 itself, which still eludes clinical definition [22]. Although some studies have evaluated the metabolome of patients during and after COVID-19 [23,24], to the best of our knowledge, no current studies have evaluated the metabolome prior to, during, and after disease diagnosis in the same study population. 

In this study, we generated comprehensive metabolomic profiles, measuring 1546 metabolites of 609 plasma samples, reflecting 555 individuals from the Longitudinal Electronic Medical Record (EMR) and Omics COVID-19 Cohort (LEOCC), a subset of the Massachusetts General Brigham Biobank. Using regression modeling, we evaluated changes in metabolite levels that are associated with COVID-19 severity and the time point at which samples were collected, being before (*n* = 441), during (*n* = 86), and after (*n* = 82) COVID-19 infection, where most patients only have a single time of sample collection (Appendix A). Models were adjusted for key demographic factors (age, sex, race, ethnicity), risk factors (body mass index (BMI), smoking status before COVID-19), and comorbidity (Charlson Index) covariables. Our study design addresses prior studies’ limitations, and serves as an important step toward the development of putative biomarkers and preventative strategies.

## 2. Results

### 2.1. Study Design

This study generated metabolomic profiles from 609 plasma samples, drawn from 555 distinct patients. Of these samples, 441 samples were collected before COVID-19 diagnosis, 86 during COVID-19 (within 28 days), and 82 after COVID-19 diagnosis (>28 days). The overall study design is in Figure 1, and information on numbers of patients with repeat or distinct samples can be found in Appendix A. Basic demographic data, risk factors, and comorbidities of our study population are shown in Table 1 (see Section 4). A wide array of chemical classes is represented in this profiling, including lipids and amino acids, the largest classes, as well as cofactors and vitamins, nucleotides, peptides, carbohydrates, and energy metabolites (Figure 2A).

Appendix A outlines the regression analyses that were conducted. “Severity Association Models”, using ordinal logistic regression with COVID-19 severity as outcome (see Section 4), aimed to identify (1) predisposition metabolites that are associated with severity prior to COVID-19 diagnosis; (2) predisposition-and-acute metabolites, reflecting predisposition metabolites that continue to be associated with severity during active disease; and (3) persistent metabolites that are associated with severity prior to, during, and after disease. All metabolites of interest relate back to pre-COVID-19 samples to ensure that interpretations were made relative to the baseline metabolomic state of the individuals.

“Time Association Models”, using linear regression with metabolites as outcome, aimed to identify (1) transient metabolites, which are altered primarily in response to active disease; and (2) lingering metabolites, which are altered with disease and show continued changes post-COVID-19. All models were adjusted for basic demographic factors (age, sex, ethnicity, race), risk factors (BMI, smoking status), and comorbidities (based on Charlson index), and FDR-adjusted *p*-values < 0.05 were considered statistically significant.

### 2.2. Metabolites Associated with Severity

We aimed to identify metabolites associated with COVID-19 disease severity for predisposition, predisposition-and-acute, and persistent metabolites (see Equation (1), Section 4, Figure 1). The number of metabolites, by class, within each and shared across these sets are depicted in Appendix A. Metabolite classes positively or negatively associated with COVID-19 severity in each of the three sets are depicted in Figure 2B (Appendix A). Overall, fewer metabolites (14%) were negatively, rather than positively, associated with COVID-19 severity, and no metabolites negatively associated with COVID-19 severity in the persistent metabolite sets mapped to pathways.

Notably, amino acid metabolites were the primary chemical class that was positively associated with severity in all three sets, while lipids dominated the negative associations with COVID-19 severity in the predisposition metabolites. This observation is in contrast to lipids comprising the largest class of metabolites measured in our platform. Nucleotides were also associated with severity in all sets. This observation suggests that nucleotide metabolism, in particular, may be associated with predisposition and continually perturbed metabolism related to COVID-19 severity.

The log odds of metabolites that were significantly associated with severity were organized by the pathway they map to in each of the three severity sets (predisposition, predisposition-and-acute, persistent). Figure 3 depicts the log odds of metabolites that mapped to clustered pathways for each set of interest (i.e., predisposition, predisposition-and-acute, and persistent metabolites), grouped by pathway cluster. Full results are available in Appendix A. Metabolites involved in nucleotide metabolism were associated with COVID-19 severity in all three sets of metabolites. Of note, none of these predisposition nucleotides were chemically modified, while all persistent nucleotides, aside from cytidine and orotidine, were either methylated, succinylated, or acetylated.

Distinct amino acid species showed specificity in the directionality of associations and/or in the metabolite sets they were significantly altered in (Figure 3). For example, some amino acids (formiminoglutamate, methylmalonate, ethylmalonate, 3-hydroxy-3-methylglutarate, aconitate, kynurenine, cystathionine, and 3-ureidopropionate) showing a predisposition to severe COVID-19 were positively associated with severity, while others (homoarginine, histidine, 4-methyl-2-oxopentanoate, and 3-methyl-2-oxobutryate) were negatively associated with severity. Further, predisposition metabolites in the amino acid-related pathways were largely related to branched-chain amino acid and arginine and proline, and sulfur-containing amino acid metabolism pathways. Metabolites that were involved in amino acid-related pathways showing predisposition to severity and acute effects were also related to sulfur-containing amino acid metabolism, in addition to tryptophan metabolism and the urea cycle. These metabolites were all positively associated with severity, except for uridine. Amino acids related to essential amino acid metabolism were also persistently and positively associated with severity across all models.

Lastly, predisposition metabolites mapped to pathways related to bile acid, stress response, and sphingolipid metabolism (Figure 3). Sphingolipids and anti-oxidant lipids related to the stress response were consistently negatively associated with COVID-19 severity, while all bile acid-related metabolites except dehydroepiandosterone sulfate were positively associated with COVID-19 severity. Sugars positively associated with predisposition and acute effects, but not persistent effects, were related to carbohydrate metabolism. Metabolites related to micronutrient and one-carbon metabolism were positively associated with severity in all time points.

Importantly, we performed a sensitivity analysis to investigate the contribution of individual comorbidities to severity associations, and resulting metabolite sets were compared to those obtained with the original models. Our results show that all but one model resulted in substantial (>73%) overlap of significant metabolites obtained from the original models (Appendix A). Only persistent metabolites from models that included CVD resulted in a lower overlap (13%) with the persistent metabolite set from the original models. We note, however, that only seven metabolites were found significant in this CVD only model, all of which overlapped with the Charlson Index model results. Thus, we interpret our results as independent of the comorbidities evaluated. Further, we verified that metabolite levels were not associated with sample collection time. In these correlation analyses, only 1 of 93 identified predisposition metabolites, 1 of 83 identified predisposition-and-acute metabolite, and 11 of 37 identified persistent metabolites were also significantly associated with collection date. Thus, potential biases due to sample collection timing is minimal in this study.

### 2.3. Metabolites Associated with Time of Sample Collection

We used the same metabolomic profiles as in the severity analysis to identify metabolites associated with time of sample collection relative to COVID-19 diagnosis (pre-, during, post-COVID-19), independent of severity (see Equation (2), Section 4), to highlight transient alterations during COVID-19 infection or lingering changes after COVID-19 infection (see Figure 1). Figure 4 depicts significant metabolites that mapped to pathways in these analyses.

#### 2.3.1. Metabolites Associated with Transient Alterations during COVID-19 Infection

Metabolites showing transient alterations in metabolism during infection are depicted in Figure 4A and listed in Appendix A. Metabolites that decreased upon infection then increased back to pre-COVID-19 levels after infection were characterized by proline-related amino acids, secondary bile acids, and metabolites of fat-soluble vitamins A and E. A larger portion of metabolites were increased upon infection and then decreased afterwards. These included metabolites that mapped to pathways related to amino acid metabolism for a variety of amino acid classes, fatty acid metabolism, glucose homeostasis, ketone body metabolism, membrane transport, nucleotide metabolism, sphingolipid metabolism, sulfur and one-carbon metabolism, and urea cycle metabolism.

#### 2.3.2. Metabolites Associated with Lingering Alterations

Metabolites associated with lingering alterations to metabolism following COVID-19 infection are depicted in Figure 4B and listed in Appendix A. Five metabolites were significantly decreased after COVID-19 infection: mannose, heme, N1-methyladenosine, N4-acetylcytidine, and thyroxine. Substantially more (26 metabolites) were increased after COVID-19 infection, including amino acids and metabolites related to nucleotide metabolism, phospholipid metabolism, porphyrin metabolism, sphingolipid metabolism, and TCA cycle metabolism.

## 3. Discussion

This is the first large comprehensive metabolomics study, comprising more than 550 individuals, that capitalized on COVID-19 plasma samples before, during, and/or after disease. Although a few other studies have leveraged similar numbers of samples [25,26,27], the samples used only reflect active and/or recovered disease states. If controls were collected before 2020, they do not necessarily reflect patients who later go on to acquire COVID-19 [27]. In our study, the multiple times of sample collection were used to capture metabolites associated with disease severity, as well as metabolomic changes that were transient during the active disease and those that lingered after the disease. We examined associations between relative metabolite abundances and COVID-19 severity at three time points (pre-, during, and post-COVID-19 infection), as well as relative differences in metabolite intensities across times of sample collection. The use of both severity and time of sample collection models enabled the identification of potential predisposing and persistent biological underpinnings of COVID-19, which may aid in developing preventive and therapeutic measures. Importantly, significant metabolite associations uncovered in this study were independent of key covariables known to be associated with COVID-19 outcomes (patient age, sex, race, ethnicity, BMI, smoking status, comorbidities, and sample collection time point). 

Globally, our study uncovered previously described as well as novel metabolites associated with COVID-19. Metabolites of interest associated with severity primarily consisted of nucleotides, amino acids, and lipids. Regression models were also used to determine metabolite perturbations during and after disease, revealing amino acid, sphingolipid and phospholipid, nucleotide, and sulfur/one-carbon metabolism pathways that were transient (only changing in intensity during COVID-19) or lingering (remaining changed after COVID-19). Approximately three quarters of metabolites (113 out of 146) were newly identified and mapped to pathways in the present study, while the remaining quarter overlapped with previous studies (Appendix A). Overall, these results provide credence to the relevance of both our findings as well as the utility of metabolomics approaches for identifying putative biomarkers of severity and understanding COVID-19 molecular underpinnings. The novelty of our investigation is two-fold: (1) predisposition metabolites, measured in pre-COVID-19 samples, were newly identified, providing evidence for pathways that potentially predispose individuals to severe disease, and (2) novel acute and persistent metabolites associated with severe disease were uncovered.

Nucleotide metabolites associated with COVID-19 severity in the three sets of interest (predisposition, predisposition-and-acute, and persistent) reflect the unique interpretations made possible by the inclusion of pre- and post-COVID-19 samples in our study design. Notably, nucleotides positively associated with only a predisposition to COVID-19 severity were unmodified, while those persistently associated with severity were largely methylated, acetylacted, or succinylated. This could potentially reflect persistent viral RNA metabolism associations with severe COVID-19, as metabolites like N4-acetylcytidine [28] and N1-methyladenosine [29] have been implicated in RNA stability. This is also consistent with a review of studies of SARS-CoV-2 epitranscriptomics, which reports N6-methyladenosine, pseudouridine, and 2′-O-methylation of viral RNA as potentially implicated in immune evasion and promotion of viral replication [30]. Our findings suggest that the epitranscriptional modification machinery may potentially be protective against severe COVID-19. Interestingly, both N4-acetylcytidine and N1-methyladenosine were also decreased in post- compared to pre-COVID-19 samples in the time association models, suggesting the lingering metabolic effects of COVID-19 on RNA metabolism.

We also note opposite relationships between severity and uridine and cytidine metabolites in predisposition-and-acute as well as persistent metabolites, which have been reported previously in active disease [13,14,31]. However, uridine is negatively associated with COVID-19 severity, even in samples collected before diagnosis, while cytosine metabolites and other upstream uridine metabolites are positively associated with COVID-19 severity, regardless of the time of sample collection. This observation suggests that the pathways of uridine metabolism and/or pyrimidine salvage could be further evaluated as potential targets in mitigating severe COVID-19, potentially in a preventive capacity. In time point association models, nucleotide metabolites related to cytosine and uracil also increased upon infection then decreased afterward. This shift after infection could reflect reduced viral reproduction, as uracil and cytosine are relevant for SARS-CoV-2 replication [31]. Globally, both severity and time association models reflect the importance of nucleotide metabolism in COVID-19 severity and potentially associated systemic metabolism at various stages.

Tryptophan metabolites have gained wide interest recently due to their association with immunosuppression and inflammation during COVID-19 [13,14,16]. In this study, tryptophan metabolites (kynurenine, kynurenate, quinolinate, and nicotinamide riboside) are positively associated with COVID-19 severity, which is consistent with numerous previous studies (see Appendix A) [13,15,17,32,33]. Importantly, because all models are adjusted for comorbidities, our reported positive relationships between tryptophan metabolites and COVID-19 severity are worth evaluating for their unique role in COVID-19, rather than only with regard to general inflammation.

Other inflammatory-related pathways, such as those involving lyso-phosphatidylcholines, reported here and previously [12,20,26], may potentially provide favorable functions of activating immune cells and influencing vascular reactivity and endothelial activation [34]. With this in mind, our reported inverse association with severity in pre-COVID-19 samples may reflect an increased capacity to fight infection, resulting in less severe disease. Further, sphingomyelins and phosphatidylcholines were inversely associated with COVID-19 severity in previous studies evaluating existing disease [10,21,26,33,35], potentially reflecting increased viral replication [12,23]. Importantly, these inverse relationships with severity were also detected in our pre-COVID-19 samples, suggesting that increased plasma sphingomyelins and phosphatidylcholines could protect against COVID-19 severity before infection. It is also worth noting that activation of the acid sphingomyelinase/ceramide system, which converts membrane sphingomyelins to ceramides, has been implicated in SARS-CoV-2 binding and infection [36]. Thus, higher abundances of sphingomyelins could possibly decrease severity through decreased conversion into ceramide and subsequent generation of ceramide rafts for ACE2 binding of SARS-CoV-2. In our study, phosphatidylcholine, sphingomyelin, and lyso-phosphatidylcholine species had lingering increased levels compared to pre-COVID samples. Thus, these phosphatidylcholines and sphingomyelin metabolism species merit continued study to evaluate their ability to prevent severe COVID-19 if targeted.

Some limitations are important to mention to ensure appropriate interpretation of the results. One such limitation is that substantially fewer samples were available during and after diagnosis, relative to prior. The low number of patients with repeat samples also prevents paired analyses. However, the numbers of during and post-COVID-19 samples still constitute a larger number than most current metabolomics studies of COVID-19, and the inclusion of all three time points enables both the evaluation of changes over time in independent individuals and the ability to carry out two major analyses (severity and time associations). Additionally, unlike other studies that may include control samples from before 2020, our pre-COVID-19 patient samples reflect a metabolic state before confirmed diagnosis. We also note that shifts in relative abundances of metabolites within a time of sample collection are likely to be observed, as in Karu et al. [37]. However, our analyses correlating metabolite abundance with sample collection date resulted in very few significant metabolites (13 out of 213 total identified metabolites), and the aim of this analysis was to look broadly at shifts from pre- to during to post-COVID-19 collection times, which was also facilitated by the large sample size available. We also were not able to perform regressions for associations with long-COVID, because only a small subset (*n* = 20) of patients in our cohort have a reported diagnostic code (U07.1). Further, samples analyzed only reflect the original SARS-CoV-2 strain, as samples were collected before emergence of additional variants [38,39] (this does, however, eliminate the need to correct for the presence of different variants in our model). Additional studies would be needed to verify the translatability of these results across different variants.

We also note the possibility that metabolite alterations identified in this study are impacted by diet or other lifestyle changes or shared across other infectious diseases. Evaluating the specificity of metabolite levels to COVID-19 would require additional investigation. To this point, a recent study did identify metabolites that were specific to acute respiratory distress syndrome patients where their disease was induced by COVID-19 compared to influenza [40]. Other studies report oral metabolites that differentiated patients with COVID-19 from those with influenza [41], and plasma metabolites that distinguish between COVID-19 and non-COVID-19 pneumonia [42,43]. We do emphasize, though, that in this study, care was taken to account for basic demographics, risk factors, and comorbidities. We further note that the systematic generation of robust clinical phenotypes is an area of ongoing research [44], the development of which will aid in the interpretability of metabolomics data that are paired with electronic medical record data.

In conclusion, our study has uncovered key pathways and metabolites of interest in terms of disease severity prevention and persistent metabolic outcomes. These pathways and metabolites could be putative or targeted biomarkers to mitigate severe COVID-19, which remains a public healthcare concern. Metabolites belonging to classes that were associated with both severity and time point association models potentially suggest persistent metabolic effects of COVID-19. Further study is warranted to determine whether such effects are recapitulated in additional investigations and if they are associated with prolonged disease outcomes or long-COVID. Overall, this study provides important data and results that can be used to identify potentially druggable metabolic pathways, which may be targeted to prevent severe disease in the ongoing pandemic.

## 4. Materials and Methods

### 4.1. Study Design and Patient Data

The Mass General Brigham (MGB) Biobank contains ~100,000 banked plasma, serum, and DNA samples from >100,000 consented patients. Electronic Medical Record (EMR) data and lifestyle, environment, and family history surveys can also be linked to the banked samples. The Longitudinal EMR and Omics COVID-19 Cohort (LEOCC) consists of a subset of individuals with prospective plasma samples from the MGB Biobank. Possible COVID-19 patients were first identified by a positive RT-PCR or antibody test. All COVID-19 diagnoses were then confirmed with a second RT-PCR test to ensure that diagnoses reflected active disease. Patients in this cohort with a positive COVID-19 diagnosis and available plasma samples prior to COVID-19 (up to 27 October 2020) were included. No additional exclusion criteria were applied. Clinical data relevant to COVID-19 infection, including clinical measures, disease diagnoses, and COVID-19 severity were also extracted from EMR data for use in statistical models. This study was approved by the Brigham and Women’s Institutional Review Board (IRB: 2014P001109).

The subset of samples used in the analysis was derived as follows. A total of 940 plasma samples from 661 individuals were collected from consented patients and were stored at –80 °C. These samples were categorized by the time point of collection relative to a positive COVID-19 diagnosis, including 474 pre-COVID-19 samples (date of collection < date of diagnosis), 282 during COVID-19 samples (collected within 28 days of diagnosis), and 182 post-COVID-19 samples (collected more than 28 days after COVID-19 diagnosis). For patients with multiple during and/or post-COVID-19 samples, only the sample collected at the date closest to diagnosis was retained for during COVID-19, and only the sample collected at the date furthest from diagnosis was retained for post-COVID-19. Patients without BMI data were also excluded from the sample sets, yielding a total of *n* = 441 pre-COVID-19, *n* = 86 during COVID-19, and *n* = 82 post-COVID-19 samples used for analysis across 555 patients. Further details about the numbers of samples from distinct patients or those from patients represented across multiple times of sample collection are given in Appendix A. Further information about the number of days that plasma samples were collected before or after COVID-19 is given in Appendix A.

The COVID-19 severity level was determined according to WHO guidelines [45] to be 0 = ambulatory mild disease (no hospitalization), 1 = hospitalized moderate disease (hospitalized without ventilation), 2 = hospitalized severe disease (hospitalized with ventilation), or 3 = death. Further demographic factors, risk factors, and comorbidity covariables were defined before COVID-19 diagnosis for all patients, as follows: age is the patient’s numerical age; race is categorical (black, other, white); ethnicity (Hispanic/non-Hispanic), sex (female/male), and smoking (yes/no) are binary; BMI is the numerical median body mass index (BMI) for each patient; and comorbidity level is a binary “mild” or “severe” factor based on a Charlson index < 5 or ≥5, respectively.

### 4.2. Charlson Index Calculation

Charlson indexes were calculated as previously described [46,47] for each patient. Indexes were calculated based on the presence of diseases and age scores as listed in Appendix A, where categories were defined by groups of ICD-9 codes and assigned a weight from 1 to 6, where more severe comorbidity categories had a higher weight [48]. The sum of the scores of comorbidity presence plus the age score yielded the Charlson index, which is negatively associated with 10-year survival. Charlson indexes were divided into two categories: an index < 5 was defined as mild and ≥5 as severe comorbidity.

### 4.3. Metabolomics Analysis

Plasma samples were sent to Metabolon (Morrisville, NC, USA) for comprehensive metabolomic profiling of polar and nonpolar metabolite classes in plasma extracts. Samples were extracted and prepared according to methods published previously [49]. Quality control (QC) samples were used to control for batch variation throughout the run, using coefficients of variation. Four previously described [50] liquid chromatography–tandem mass spectrometry (LC-MS/MS) methods were used to capture comprehensive metabolite classes, including (1) amines and polar metabolites (positive ionization), (2) polar metabolites in central metabolism (negative ionization), (3) polar and nonpolar lipids, and (4) lipids and metabolites of intermediate polarity (including free fatty acids and bile acids). Instruments were calibrated each day for mass resolution and mass accuracy, and all materials were purchased from a single lot [51].

Raw metabolite integrals were normalized to the median value (median value = 1) of each metabolite for a given run day. Metabolite identification was performed using automated comparison to features (including retention time, *m/z*, preferred adducts, in-source fragments, and mass spectra) of ~8000 purified chemical standards in the Metabolon reference library [51]. Metabolites that remained unknown, but are reproducibly captured, after comparison with standards were annotated with a unique and persistent “X-number” for potential future identification.

Abundances for a total of 1546 grouped and aligned metabolites were reported on the Metabolon platform. Metabolites annotated as xenobiotics and with ≥75% missing values were removed, yielding 1108 metabolites in the final dataset. Remaining missing values were imputed as the half-minimum intensity for a given metabolite. Metabolite intensities were log-transformed and pareto-scaled for use in regression models.

### 4.4. Statistical Models

Regression models were applied to assess associations between metabolite levels and (1) COVID-19 severity, where samples were stratified by sample collection time point, and (2) time of sample collection (e.g., pre-, during, post-COVID-19). Models were adjusted for demographics (age, race, ethnicity, sex), risk factors (BMI, smoking status), and comorbidities (Charlson Index category). Relevant coefficient *p*-values were adjusted for multiple comparisons using false discovery rates [52], and FDR-adjusted *p*-values < 0.05 were considered statistically significant.

Equation (1) was used to determine metabolite associations with severity, stratified by pre-, during, and post-COVID-19 samples:(1)$$\begin{array}{cc} \log\left(\frac{\mathrm{Pr}\left(Y\leq j\right)}{1-\mathrm{Pr}\left(Y\leq j\right)}\right) & \\ & =\beta_{0}+\beta_{1}metabolite+\beta_{2}age+\beta_{3}race+\beta_{4}ethnicity+\beta_{5}sex+\beta_{6}BMI{+\beta }_{7}smoking\\ & +\beta_{8}comorbidity\;level \end{array}$$
where the outcome is the ordinal COVID-19 severity (levels 0–3), and metabolite is normalized and scaled relative metabolite abundance. Modeling was performed using the polr() function of the MASS R package (version 7.3-55) [53]. From these models, we determined three sets of interest: (1) predisposition metabolites, significantly associated with severity in pre-COVID-19 samples only; (2) predisposition-and-acute metabolites, significantly associated with severity in the same direction in pre- and during COVID-19 samples; and (3) persistent metabolites, significantly associated with severity in the same direction across all times of sample collection (an UpSet plot ([54]) of all significant metabolites at each time of sample collection is given in Appendix A).

We also performed a sensitivity analysis wherein we tested the effects of individual comorbidities in the models one at a time. Specifically, we replaced the comorbidity level term of Equation (1) with a binary variable indicating the presence/absence of the following comorbidities at COVID-19 diagnosis: stroke, cardiovascular disease (CVD), type 2 diabetes, chronic obstructive pulmonary disease, major depression, and cancer.

Lastly, we calculated Pearson’s correlations between significant metabolite abundances (from Equation (1)) and date of sample collection to ensure that associations with severity were not inadvertently associated with collection date.

Equation (2) was used to determine metabolites associated with time of sample collection (pre, during, and post-COVID-19):(2)$$\begin{array}{c} E\left(Y\right)=\beta_{0}+\beta_{1}time\;point+\beta_{2}severity+\beta_{3}age+\beta_{4}race+\beta_{5}ethnicity+\beta_{6}sex+\beta_{7}BMI+\beta_{8}smoking\\ \;+\beta_{9}comorbidity\;level+\varepsilon \end{array}$$
using the glm() function in R (version 4.1.3), where E(Y) is the normalized and scaled metabolite relative abundance, time point is the time of sample collection (pre-, during, and post-COVID-19), and ε follows a normal distribution N(0,1).

The results of Equation (2) were used to identify metabolites that reflect transient alterations during infection and lingering changes after infection. Specifically, we identified the following sets of metabolites based on the statistical significance of the sample collection time term: (1) those altered upon active infection (pre- vs. during COVID-19), (2) metabolites altered after active infection relative to pre-COVID-19 samples (pre- vs. post-COVID-19 samples), and (3) metabolites altered after infection relative to active infection (during vs. post-COVID-19). From these significant metabolites, we defined transient and lingering alterations as follows: (1) transient alterations during infection include metabolites significantly associated in pre- vs. during COVID-19 and in during vs. post-COVID-19, but not pre- vs. post-COVID-19; (2) lingering alterations after infection include metabolites significantly associated in pre- vs. post-COVID-19.

### 4.5. Mapping Metabolites to Pathways

Statistically significant metabolites were mapped to pathways using the getPathwayFromAnalyte() function from RaMP-DB 2.0 (database version 2.0.7, package version 2.3.1) [55] to assist with biological interpretation. Metabolite identifiers (HMDB, PubChem, CAS, and ChemSpider) provided by the Metabolon platform and the custom background of all identified metabolites in the entire metabolomics dataset (*n* = 1276) were used as input. Only pathways that contained at least two metabolites were retained. Pathways that shared common metabolites/annotations were initially clustered using the findCluster() function. Clusters were then manually inspected and annotated to revise cluster membership (see Appendix A for mapping and Appendix A for metabolites that mapped to unclustered pathways).

## Figures and Tables

**Figure 1 ijms-25-00346-f001:**
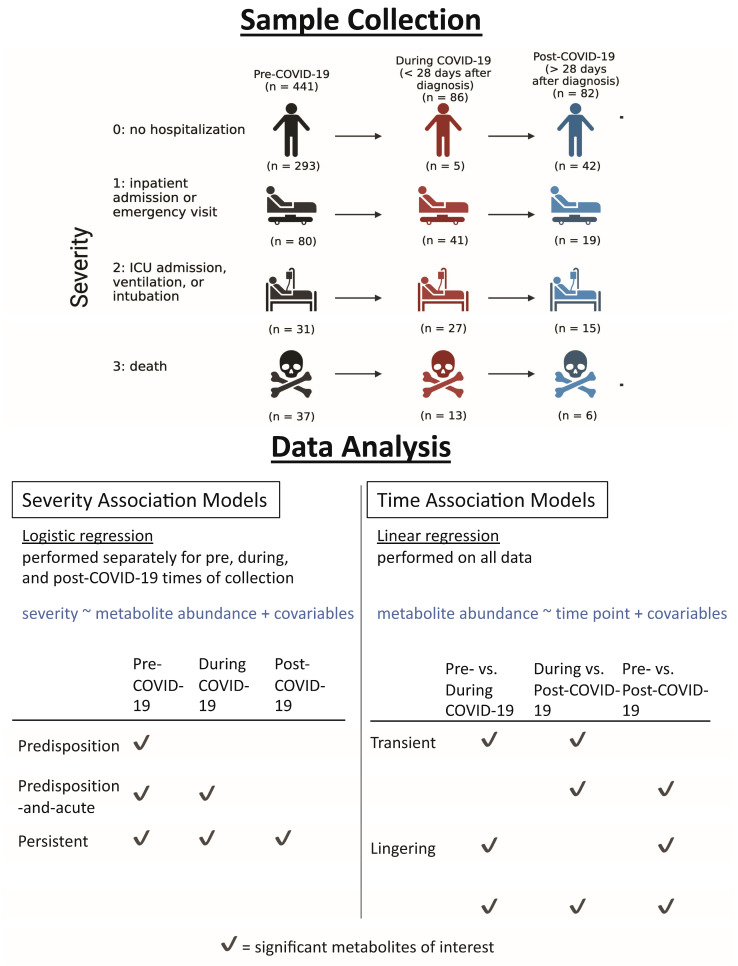
Study design schematic. Metabolomic profiles of plasma samples drawn from 555 COVID-19 patients were analyzed. The majority of samples reflect pre-COVID-19 timepoints (441), while 86 were drawn during and 82 post-COVID-19 diagnosis (only 5 patients had samples available at all three timepoints, see Appendix A). Severity levels ranged from 0 to 3, according to WHO guidelines. Statistically significant metabolites in each analysis were identified as those with FDR-adjusted *p*-values < 0.05. Metabolite associations with COVID-19 severity were determined using Equation (1), stratified by each time of sample collection. Sets of interest are described in the table, where checkmarks indicate that metabolites must be significant and share directionality of association for that comparison to be included in the same set. Metabolite associations with time of sample collection, Equation (2), were determined using all time points. Checkmarks for these associations indicate significant metabolites in each comparison.

**Figure 2 ijms-25-00346-f002:**
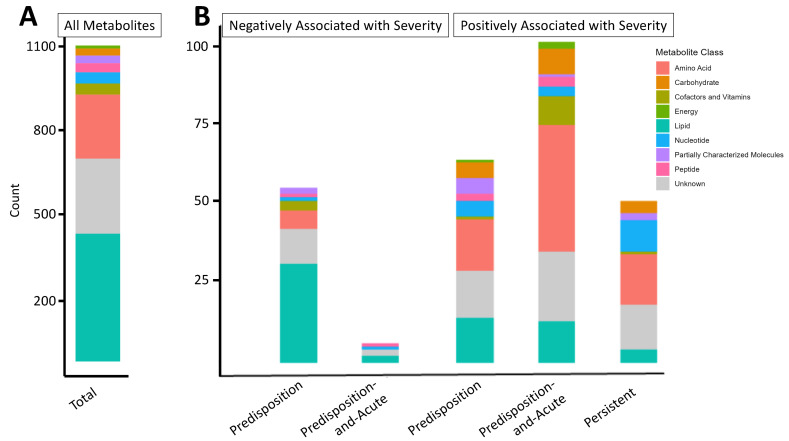
Metabolite classes measured and associated with COVID-19 severity. (**A**) Representation of total metabolite classes detected by the Metabolon comprehensive metabolomics platform used in regression modeling (lipids = 447, unknown = 264, amino acids = 225, cofactors and vitamins = 39, nucleotides = 39, peptides = 32, partially characterized molecules = 27, carbohydrates = 25, energy = 10). (**B**) Metabolite classes for metabolites significantly associated with severity (FDR-adjusted *p*-value < 0.05; for the severity term, see Equation (1)) that fall into our three metabolite sets of interest: (1) predisposition metabolites, identified from pre-COVID-19 samples only; (2) predisposition-and-acute metabolites, defined as those significant and with the same directionality of association with severity in pre- and during COVID-19 samples; and (3) persistently affected metabolites, defined as those significant and with the same directionality of association with severity in pre-, during, and post-COVID-19 samples.

**Figure 3 ijms-25-00346-f003:**
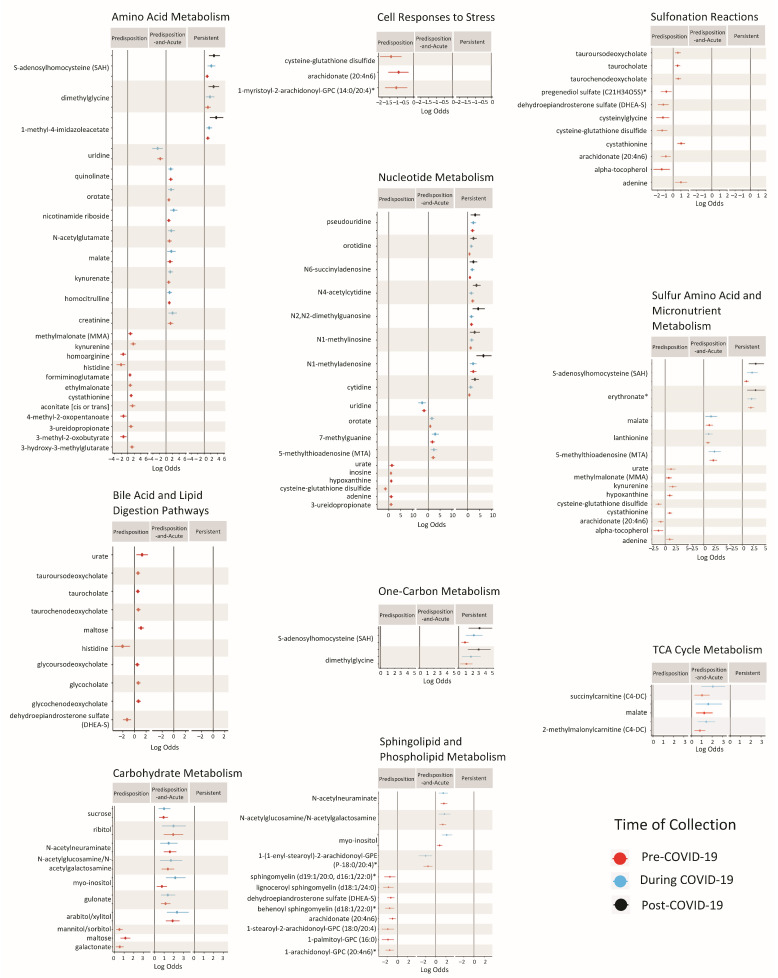
Metabolites significantly associated with COVID-19 severity associations that map to biological pathways. Each panel represents a pathway cluster (group of similar pathways). Log odds of significant associations between metabolites and COVID-19 severity (Equation (1)). Sets of interest include predisposition metabolites (only significantly associated with severity (FDR-adjusted *p*-values < 0.05) in pre-COVID-19 samples (*n* = 441)) (see Appendix A); predisposition-and-acute metabolites (significantly associated with severity (FDR-adjusted *p*-values < 0.05) in pre- (*n* = 441) and during (*n* = 86) COVID-19 samples) (see Appendix A); and persistent metabolites (significantly associated with severity (FDR-adjusted *p*-values < 0.05) in all samples (*n* = 441, 86, and 82 for pre-, during, and post-COVID-19 samples, respectively) (see Appendix A). Metabolites can appear in multiple pathway clusters because some metabolites map to multiple pathways. * Identities of these metabolites are confidently assigned by Metabolon but without authentic standards.

**Figure 4 ijms-25-00346-f004:**
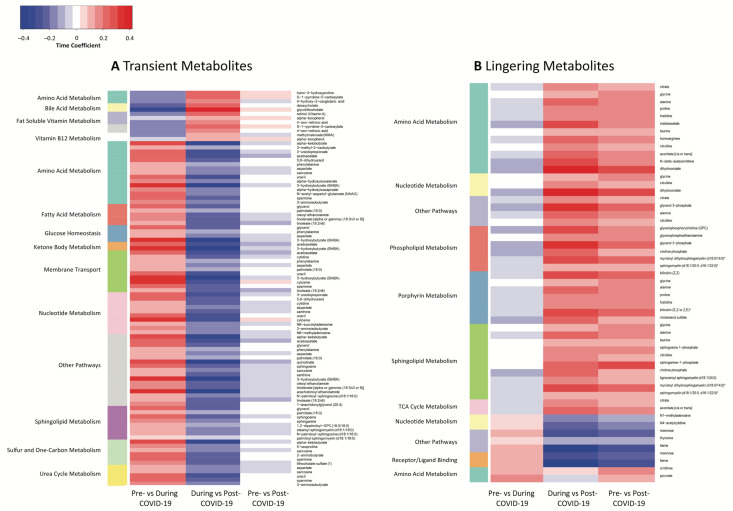
Metabolite abundance changes over time in samples collected prior to, during, and after COVID-19 for metabolites that map to biological pathways. Change in effect sizes (coefficient of the time term of Equation (2) where the change is relative to the latter group of the compared pairs) of metabolites that reflect (**A**) transient changes during and (**B**) lingering changes after COVID-19 infection. For (**A**), metabolites were significantly associated with time in pre- (*n* = 441) vs. during (*n* = 86) COVID-19 and during vs. post-COVID-19 (*n* = 82) (FDR-adjusted *p*-value < 0.05) but not pre- vs. post-COVID-19 comparisons, and mapped to pathways using RaMP-DB. For (**B**), metabolites were significantly associated with time in pre- vs. post-COVID-19 (FDR-adjusted *p*-value < 0.05) comparisons and mapped to pathways using RaMP-DB. See also Appendix A for FDR-adjusted *p*-values.

**Table 1 ijms-25-00346-t001:** Cohort demographics of individuals with plasma samples collected pre-, during, and post-COVID-19 diagnosis. Severity levels (0–3) are based on the WHO COVID-19 severity scale where 0 = no hospitalization, 1 = emergency room visit, 2 = intubation/ICU visit, 3 = death. Continuous covariables are represented as mean (standard deviation), and categorical covariables are represented as number of samples (percentage within time and severity level stratum).

	Pre-COVID-19					During COVID-19					Post-COVID-19				
	0	1	2	3	Total	0	1	2	3	Total	0	1	2	3	Total
Number of samples	293	80	31	37	441	5	41	27	13	86	42	19	15	6	82
Demographic Covariables															
Age	46.6 (15.0)	54.4 (16.3)	54.8 (13.2)	66.8 (16.2)	50.3 (16.2)	58.4 (27.3)	57.7 (16.8)	62.7 (11.3)	62.8 (13.1)	60.1 (15.4)	48.4 (15.4)	52.7 (18.5)	59.5 (11.6)	65.0 (17.6)	52.6 (16.4)
Sex															
Female	199 (67.9%)	47 (58.8%)	15 (48.4%)	9 (24.3%)	270 (61.2%)	4 (80.0%)	25 (61.0%)	16 (59.3%)	8 (61.5%)	53 (61.6%)	31 (73.8%)	12 (63.2%)	8 (53.3%)	3 (50.0%)	54 (65.9%)
Male	94 (32.1%)	33 (41.2%)	16 (51.6%)	28 (75.7%)	171 (38.8%)	1 (20.0%)	16 (39.0%)	11 (40.7%)	5 (38.5%)	33 (38.4%)	11 (26.2%)	7 (36.8%)	7 (46.7%)	3 (50.0%)	28 (34.1%)
Race															
Black	40 (13.7%)	12 (15.0%)	2 (9.7%)	4 (10.8%)	59 (13.4%)	2 (40.0%)	16 (39.0%)	7 (25.9%)	4 (30.8%)	29 (33.7%)	2 (4.8%)	6 (31.6%)	2 (13.3%)	1 (16.7%)	11 (13.4%)
White	162 (55.3%)	53 (66.3%)	20 (64.5%)	29 (78.4%)	264 (59.9%)	1 (20.0%)	16 (39.0%)	15 (55.6%)	7 (53.8%)	39 (45.3%)	35 (83.3%)	8 (42.1%)	9 (60.0%)	5 (83.3%)	57 (69.5%)
Other	91 (31.1%)	15 (18.8%)	8 (25.8%)	4 (10.8%)	118 (26.8%)	2 (40.0%)	9 (22.0%)	5 (18.5%)	2 (15.4%)	18 (20.9%)	5 (11.9%)	5 (26.3%)	4 (26.7%)	0 (0%)	14 (17.1%)
Ethnicity															
Hispanic	58 (19.8%)	15 (18.8%)	6 (19.4%)	4 (10.8%)	83 (18.8%)	2 (40.0%)	5 (12.2%)	5 (18.5%)	2 (15.4%)	14 (16.3%)	2 (4.8%)	2 (10.5%)	2 (13.3%)	0 (0%)	6 (7.3%)
Non-Hispanic	235 (80.2%)	65 (81.3%)	25 (80.6%)	33 (89.2%)	358 (81.2%)	3 (60.0%)	36 (87.8%)	22 (81.5%)	11 (84.6%)	72 (83.7%)	40 (95.2%)	17 (89.5%)	13 (86.7%)	6 (100%)	76 (92.7%)
Risk Factor Covariables															
Smoking Status															
Yes	92 (31.4%)	42 (52.5%)	15 (48.4%)	27 (73.0%)	176 (39.9%)	1 (20.0%)	22 (53.7%)	10 (37.0%)	5 (38.5%)	38 (44.2%)	10 (23.8%)	9 (47.4%)	5 (33.3%)	2 (33.3%)	26 (31.7%)
No	201 (68.6%)	38 (47.5%)	16 (51.6%)	10 (27.0%)	265 (60.1%)	4 (80.0%)	19 (46.3%)	17 (63%)	8 (61.5%)	48 (55.8%)	32 (76.2%)	10 (52.6%)	10 (66.7%)	4 (66.7%)	56 (68.3%)
BMI	30.8 (6.39)	30.6 (5.84)	31.6 (7.07)	30.7 (5.40)	30.8 (6.25)	34.3 (5.63)	31.4 (6.60)	34.2 (7.59)	32.5 (9.64)	32.6 (7.38)	29.8 (7.73)	30.7 (6.54)	34.5 (6.33)	36.2 (6.89)	31.4 (7.38)
Comorbidity Covariable															
Charlson Index Category															
Mild	152 (51.9%)	19 (23.8%)	5 (16.1%)	1 (2.7%)	177 (40.1%)	3 (60.0%)	11 (26.8%)	8 (29.6%)	1 (7.7%)	23 (26.7%)	30 (71.4%)	8 (42.1%)	7 (46.7%)	0 (0%)	45 (54.9%)
Severe	141 (48.1%)	61 (76.3%)	26 (83.9%)	36 (97.3%)	264 (59.9%)	2 (40.0%)	30 (73.2%)	19 (70.4%)	12 (92.3%)	63 (73.3%)	12 (28.6%)	11 (57.9%)	8 (53.3%)	6 (100%)	37 (45.1%)

## Data Availability

Metabolome data, processed and normalized as described in the Methods, and accompanying annotations are publicly available on Metabolomics Workbench (study ID: ST002829) [56]. Metadata cannot be made public to maintain privacy of patients included in the study; however, metadata can be made available by the corresponding author upon request.

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
