# Peer review of "Nucleotide, Phospholipid, and Kynurenine Metabolites Are Robustly Associated with COVID-19 Severity and Time of Plasma Sample Collection in a Prospective Cohort Study"

_ijms, 2023, doi:10.3390/ijms25010346_

Round 1

Reviewer 1 Report

Comments and Suggestions for Authors

The manuscript presents novel and important data on the study of metabolomics in COVID-19 patients.

Here are some questions and suggestions for the improvement of the manuscript:

1. It must be stated early in the abstract that pre, during, and post COVID-19 diagnosis samples are not all coming from the same patients. This is an important limitation that needs to be clear early in the paper.

2. Kindly modify Figure 1 to improve presentation of study design. Current format is too crowded and difficult to understand without going through the caption in detail, which defeats the purpose of a summative figure.

3. Table 1 needs to be improved since current format is difficult to understand. Also, it is suggested to be clear about the data regarding which patients have only pre COVID-19 samples, etc.

4. Please clarify how many samples and distinct individuals were eventually included in the metabolite analysis.

Author Response

Reviewer 1

The manuscript presents novel and important data on the study of metabolomics in COVID-19 patients.

We thank the reviewer for their kind comments and added insight. The comments have made the paper stronger and methods clearer.

Here are some questions and suggestions for the improvement of the manuscript:

  1. It must be stated early in the abstract that pre, during, and post COVID-19 diagnosis samples are not all coming from the same patients. This is an important limitation that needs to be clear early in the paper.

We agree this is indeed an important distinction to state up-front. We have updated the abstract to clarify that most patients have single timepoints. We hope that this clarifies, in the abstract, that not all patients had repeat sampling.

  1. Kindly modify Figure 1 to improve presentation of study design. Current format is too crowded and difficult to understand without going through the caption in detail, which defeats the purpose of a summative figure.

We have reorganized Figure 1 to more clearly depict the study design. We separated out the sample collection and data analysis schemes and moved the sample availability UpSet plot to a new supplementary figure (S1).

  1. Table 1 needs to be improved since current format is difficult to understand. Also, it is suggested to be clear about the data regarding which patients have only pre COVID-19 samples, etc.

We have updated Table 1 to more clearly depict the distribution of patient demographics. We removed the “All Time Points” column and instead organized the table with columns representing the time of sample collection and severity level. We hope this more clearly reflects the distributions of covariables represented in the models and simplifies understanding.

  1. Please clarify how many samples and distinct individuals were eventually included in the metabolite analysis.

As with the abstract, we agree that it is important to note the number of distinct patients. We have clarified the number of distinct individuals and samples used in the models in the following sections (line numbers reflect the numbers when “All Markup” is selected for tracking changes):

  • Introduction: lines 84, 89-90
  • Results: lines 97-98, 102-104
  • Figure 1 caption: line 111-113
  • Discussion: line 266
  • Methods: line 414 (clarifying the subsetting of samples), 425-427

Reviewer 2 Report

Comments and Suggestions for Authors

 The way this study is introduced is misleading: the authors claim to have a very large cohort of subjects sampled before, during and after COVID-19 infection. For sure this type of multiple sampling would be particularly valuable if conducted on the same subjects. On this point the text is unclear.

Indeed, at lines 403-405, one reads the following numbers:

441 samples were collected before COVID-19 diagnosis

86 during COVID-19 (within 28 days)

82 after COVID-19 diagnosis (>28 days)

As more than 600 subjects (line 258) were involved, one is led to conclude that the disease model is only built on 86 individuals. And that the pre- and post-COVID were independent subjects (441+86+82=609).

At lines 396-401, different figures are provided: “A total of 940 plasma samples from 661 individuals were collected from consented patients and were stored at –80 C. These samples are categorized by the time point of collection relative to a positive COVID-19 diagnosis, including 474 pre-COVID-19 samples (date of collection < date of diagnosis), 282 during COVID-19 samples (collected within 28 days of diagnosis), and 182 post-COVID-19 samples (collected more than 28 days after COVID-19 diagnosis). “

The authors shall clarify the above aspect.

They should also smooth some of their statements (including at line 258 “This is the first large comprehensive metabolomics study…” and cite the appropriate literature. For example:

·       I disagree with the following sentence of the abstract: “This is the first large metabolomics study using COVID-19 plasma samples before, during, and/or after disease.” The authors should look at the existing literature (e.g., papers by J. Nicholson, O, Millet, P. Turano).

·       Ref 10 could be integrated with a plethora of articles dealing with alteration in lipids and lipoproteins (including those by the authors listed above).

·       Lines 61-64: as for the abstract, the authors do not demonstrate a comprehensive view of existing literature. For example, when commenting about number of patients:

Metabolites, 2022 Dec 1;12(12):1206                           ~900 patients + 513 post-COVID

PLoS Pathog, 2023 Nov 9;19(11):e1011787                  >500 patients of which ~150 re-sampled post-COVID + 95 independent subjects post-COVID

What is the meaning of having 26 metabolites increased after COVID? It is not clear whether they are related to long-COVID effects. Is there a clinical evaluation for the individuals sampled post-COVID that could confirm of exclude a diagnosis of long-COVID?

Author Response

Reviewer 2

The way this study is introduced is misleading: the authors claim to have a very large cohort of subjects sampled before, during and after COVID-19 infection. For sure this type of multiple sampling would be particularly valuable if conducted on the same subjects. On this point the text is unclear. 

We thank the reviewer for the careful consideration of our paper and study design. We have addressed the comments, below, and believe they have made the paper stronger.

Indeed, at lines 403-405, one reads the following numbers:

441 samples were collected before COVID-19 diagnosis

86 during COVID-19 (within 28 days)

82 after COVID-19 diagnosis (>28 days)

As more than 600 subjects (line 258) were involved, one is led to conclude that the disease model is only built on 86 individuals. And that the pre- and post-COVID were independent subjects (441+86+82=609).

At lines 396-401, different figures are provided: “A total of 940 plasma samples from 661 individuals were collected from consented patients and were stored at –80 C. These samples are categorized by the time point of collection relative to a positive COVID-19 diagnosis, including 474 pre-COVID-19 samples (date of collection < date of diagnosis), 282 during COVID-19 samples (collected within 28 days of diagnosis), and 182 post-COVID-19 samples (collected more than 28 days after COVID-19 diagnosis). “

The authors shall clarify the above aspect.

We agree that it is important to note the number of distinct patients. We have clarified the number of distinct individuals and samples used in the models in the following sections (line numbers reflect the numbers when “All Markup” is selected for tracking changes):

  • Introduction: lines 84, 89-90
  • Results: lines 97-98, 102-104
  • Figure 1 caption: line 111-113
  • Discussion: line 266
  • Methods: line 414 (clarifying the subsetting of samples), 425-427

They should also smooth some of their statements (including at line 258 “This is the first large comprehensive metabolomics study…” and cite the appropriate literature. For example:

  • I disagree with the following sentence of the abstract: “This is the first large metabolomics study using COVID-19 plasma samples before, during, and/or after disease.”The authors should look at the existing literature (e.g., papers by J. Nicholson, O, Millet, P. Turano).

      We have included references to these papers in our discussion (lines 267-270, where line numbers are those when “All Markup” is selected for track changes) and in our Supplementary File 1. We note, however, that these papers do have large cohorts, but they are limited to acute and post-COVID-19 time points. If they do include pre-COVID-19 samples (e.g., Lodge et al. has an “HC” control cohort from before 2020), they do not include information about whether these patients eventually contract COVID-19. We have clarified this in lines 268-270 and we hope that the new structure facilitates understanding of the uniqueness of our study.

  • Ref 10 could be integrated with a plethoraof articles dealing with alteration in lipids and lipoproteins (including those by the authors listed above).

      We have included additional articles with ref 10.

  • Lines 61-64: as for the abstract, the authors do not demonstrate a comprehensive view of existing literature. For example, when commenting about number of patients:

Metabolites, 2022 Dec 1;12(12):1206                           ~900 patients + 513 post-COVID

PLoS Pathog, 2023 Nov 9;19(11):e1011787                  >500 patients of which ~150 re-sampled post-COVID + 95 independent subjects post-COVID

We have included updated citations based on more recent literature. This is also reflected in additional studies cited in Supplementary File 1.

What is the meaning of having 26 metabolites increased after COVID? It is not clear whether they are related to long-COVID effects. Is there a clinical evaluation for the individuals sampled post-COVID that could confirm of exclude a diagnosis of long-COVID?

We considered including an analysis for long-COVID upon seeing these results. Unfortunately, only 20 patients in our cohort have and ICD-10CM diagnosis for long-COVID, which precludes a clinical evaluation of these metabolites in long-COVID.

Round 2

Reviewer 1 Report

Comments and Suggestions for Authors

No further comments.

Reviewer 2 Report

Comments and Suggestions for Authors

Acceptable in the present form